# Quantitative detection of sleep apnea with wearable watch device

**Junichiro Hayano**[1]*, **Hiroaki Yamamoto**[2], **Izumi Nonaka**[2], **Makoto Komazawa**[3], **Kenichi Itao**[3], **Norihiro Ueda**[1], **Haruhito Tanaka**[2], **Emi Yuda**[4]

**1** Nagoya City University Graduate School of Medical Sciences, Nagoya, Japan, **2** Gifu Mates Sleep Clinic, Gifu, Japan, **3** WINFrontier Co., Ltd., Tokyo, Japan, **4** Tohoku University Graduate School of Engineering, Sendai, Japan

* hayano@acm.org

**Data Availability Statement:** The data contain potentially identifying or sensitive patient information and cannot be shared publicly. The use of the data is limited to the purpose and method of research approved by the Research Ethics

## Abstract

The spread of wearable watch devices with photoplethysmography (PPG) sensors has made it possible to use continuous pulse wave data during daily life. We examined if PPG pulse wave data can be used to detect sleep apnea, a common but underdiagnosed health problem associated with impaired quality of life and increased cardiovascular risk. In 41 patients undergoing diagnostic polysomnography (PSG) for sleep apnea, PPG was recorded simultaneously with a wearable watch device. The pulse interval data were analyzed by an automated algorithm called auto-correlated wave detection with adaptive threshold (ACAT) which was developed for electrocardiogram (ECG) to detect the cyclic variation of heart rate (CVHR), a characteristic heart rate pattern accompanying sleep apnea episodes. The median (IQR) apnea-hypopnea index (AHI) was 17.2 (4.4–28.4) and 22 (54%) subjects had AHI $\geq$15. The hourly frequency of CVHR (Fcv) detected by the ACAT algorithm closely correlated with AHI ($r = 0.81$), while none of the time-domain, frequency-domain, or non-linear indices of pulse interval variability showed significant correlation. The Fcv was greater in subjects with AHI $\geq$15 (19.6 ± 12.3 /h) than in those with AHI <15 (6.4 ± 4.6 /h), and was able to discriminate them with 82% sensitivity, 89% specificity, and 85% accuracy. The classification performance was comparable to that obtained when the ACAT algorithm was applied to ECG R-R intervals during the PSG. The analysis of wearable watch PPG by the ACAT algorithm could be used for the quantitative screening of sleep apnea.

## Introduction

Sleep apnea is a common health problem affecting 10–30% of adults [1, 2] and is associated with sleepiness, reduced sleep quality, reduced labor and learning capacity, frequent traffic accidents [3, 4], and increased cardiovascular disease risk [5]. Due to the limited clinical resources of standard sleep apnea testing by polysomnography (PSG), a variety of simple test devices have been developed to screen for sleep apnea at home [6, 7]. However, many patients may not have the opportunity to be tested, because most patients with sleep apnea don't have

Committee of the Graduate School of Medicine, Nagoya City University and Nagoya City University Hospital (approval number 60-20-0004). However, other researchers may send data access requests to irb_jimu@med.nagoya-cu.ac.jp or the corresponding author (address: hayano@acm. org).

**Funding:** This study is partly funded by WINFrontier Co., Ltd. This company participated in the study design and supported the data collection, but did not have any additional role in the data analysis, decision to publish, or preparation of the manuscript. The company also provided support in the form of salaries for two of the authors (M.K. and K.I.). The specific roles of these authors are articulated in the 'author contributions' section.

**Competing interests:** This study was partly funded by WINFrontier Co., Ltd., and the company provided support in the form of salaries for two of the authors (M.K. and K.I.). This does not alter our adherence to PLOS ONE policies on sharing data. The materials, equipment, software, consumables, and systems used in this study do not include the products or services of this company. None of the other authors has any competing interests to declare.

strong subjective symptoms or awareness to motivate them to visit a clinic or to access such devices. Recently, wearable watch devices with photoplethysmography (PPG) sensors have become popular. Although most of such devices still do not provide stable pulse wave signals during daily physical activities, they can generally deliver reliable signals during sleeping [8–10]. If sleep apnea could be screened by the pulse wave signals acquired by these devices, it could be a considerable solution for this situation.

In this study, we examined if sleep apnea can be detected from PPG signals by applying an automated algorithm that has been developed for sleep apnea detection from R-R intervals of electrocardiography (ECG). The algorithm is called auto-correlated wave detection with adaptive threshold (ACAT) [11–13]. It detects the cyclic variations of heart rate (CVHR), a characteristic pattern of R-R interval variations accompanying sleep apnea episodes [14]. In a previous study of 862 subjects undergoing polysomnographic examination, the hourly frequency of CVHR detected by the ACAT algorithm showed a correlation coefficient of 0.84 with the apnea-hypopnea index (AHI) and detected subjects with AHI $\geq$15 with 83% sensitivity and 88% specificity [11]. Although studies of pulse interval (PI) variability have reported important differences in the amplitude of short-term fluctuation components from those of R-R interval (RRI) variability [15–20], ACAT algorithm may be robust to such differences because the cycle length of CVHR is long (25–130 s) and ACAT has ability to adapt the detection threshold according to the changes in CVHR amplitude.

## Materials and methods

### Ethics approval and consent to participate

This study was performed according to the protocol that has been approved by the Research Ethics Committee of Nagoya City University Graduate School of Medical Sciences and Nagoya City University Hospital (No. 60-20-0004). All subjects participated in this study gave their written informed consent.

### Subjects

The subjects of this study were consecutive patients who underwent a diagnostic overnight PSG for sleep disordered breathing at Gifu Mates Sleep Clinic (Gifu, Japan) between March 2020 and May 2020. The inclusion criterion was adults of age $\geq$20 years. Subjects were excluded if he or she had continuous atrial fibrillation, acute illness or chronic disease exacerbation requiring hospitalization within the last 3 months, or were pregnant or breastfeeding.

### Protocol

Subjects visited a sleep clinic in the evening and spent overnight in a PSG testing chamber equipped with an Alice diagnostic sleep system (Philips Respironics, Murrysville, PA, USA). During the PSG, they wore a wearable watch device (E4 wristband, Empatica, Milano, Italy) on their left wrist and made a continuous recording of the PPG.

### Measurements

The PSG examination was started at 20:00 h and the data were collected from 21:00 h to 06:00 h the next morning. The standard PSG montages consisting of F4-M1, F4-M2, C4-M1, C3-M2, O2-M1, and O1-M2 electroencephalograms, left and right electrooculograms, a submental electromyogram, a nasal pressure cannula, oronasal airflows, left and right tibial electromyograms, thoracoabdominal inductance plethysmograms, pulse oxy-metric arterial blood

oxygen saturation (SpO2), a neck microphone, body position sensors, and a modified lead II ECG.

Sleep stages and respiratory events were scored according to the AASM Manual for the Scoring of Sleep and Associated Events [21] by registered polysomnogram technicians. The average hourly frequencies of apneic episodes, hypopneic episodes, and the combination were defined as apneic index (AI), hypopneic index (HI), and apnea-hypopnea index (AHI), respectively. The average hourly frequencies of apneic episodes were also measured by the types (obstructive, central, and mixed). Additionally, the averages of hourly frequency of 2% and 3% oxygen desaturations were also measured. In the present study, time in bed instead of total sleep time was used as the denominator in the calculation of these indices. Subjects with AHI between 5 and 15 were defined as mild, those with AHI between 15 and 30 as moderate, and those with AHI ≥30 as severe sleep apnea.

The ECG signal of the polygraph was sampled at a frequency of 100 Hz for the entire length of the PSG. All QRS complexes were identified and labeled as normal (sinus rhythm), ventricular ectopic, supraventricular ectopic, and artifact, and R-R interval time series were generated using only consecutive sinus rhythm R waves.

The wearable watch device (E4 wristband) emitted green light and recorded PPG as the inverted intensity of reflected light at a sampling frequency of 64 Hz and a resolution of 0.9 nW/digit. The PPG data were uploaded offline to the manufacturer's cloud via the Internet (E4 connect, Empatica, Milan, Italy), where the PI time series were measured as the foot-to-foot intervals of the pulse waves with motion artifacts removed [22, 23].

## Data analysis

Both ECG RRI and PPG PI time series were analyzed by the same ACAT algorithm. The detail of this algorithm has been reported previously [11, 12]. Briefly, the ACAT algorithm is a time-domain method to detect the CVHR as cyclic and autocorrelated dips in beat interval time series and determines the temporal position of the dips comprising the CVHR (Figs 1 and 2). The ACAT algorithm are comprised of the following steps: First, the interval time series are smoothed by second-order polynomial fitting, and all dips with widths between 10 and 120 s and depth-to-width ratios of >0.7 ms/s are detected. Also, the upper and lower envelopes of the interval variations are calculated as the 95th and 5th percentile points, respectively, within a moving window with a width of 130 s. Second, the dips that met the following criteria are considered CVHR: (1) a relative dip depth >40% of the envelope range at the point of dip (adaptive threshold), (2) inter-dip intervals (cycle length) between 25 and 130 s, (3) a wave-form similar to those of the two preceding and two subsequent dips with a mean morphological correlation coefficients >0.4 (autocorrelated wave), and (4) three cycle lengths between four consecutive dips that meet the following equivalence criteria: $(3-2l_1/s)$ $(3-2l_2/s)$ $(3-2l_3/s)$ >0.8, where $l_1$, $l_2$, and $l_3$ are three consecutive cycle lengths and $s = (l_1+l_2+l_3)/3$. Finally, the number of dips comprising the CVHR is counted and the mean hourly frequency of the dips is calculated as Fcv.

To compare the performance of Fcv to estimate AHI, the following time-domain, frequency-domain, and nonlinear indices of pulse rate variability were also computed from both PPG PI and ECG RRI time series: the standard deviation of normal-to-normal (N-N) intervals (SDNN), the standard deviation of 5-min average N-N interval (SDANN), the root mean square of successive differences in N-N interval (rMSSD), deceleration capacity of N-N intervals (DC) [24], short-term (4–11 beats) and long-term (>11 beats) scaling exponents computed by detrended fluctuation analysis ($\alpha_1$ and $\alpha_2$) of N-N intervals [25], and the spectral exponent ($\beta$), the total power, ultra-low-frequency (<0.0033 Hz) power (ULF), very-low-

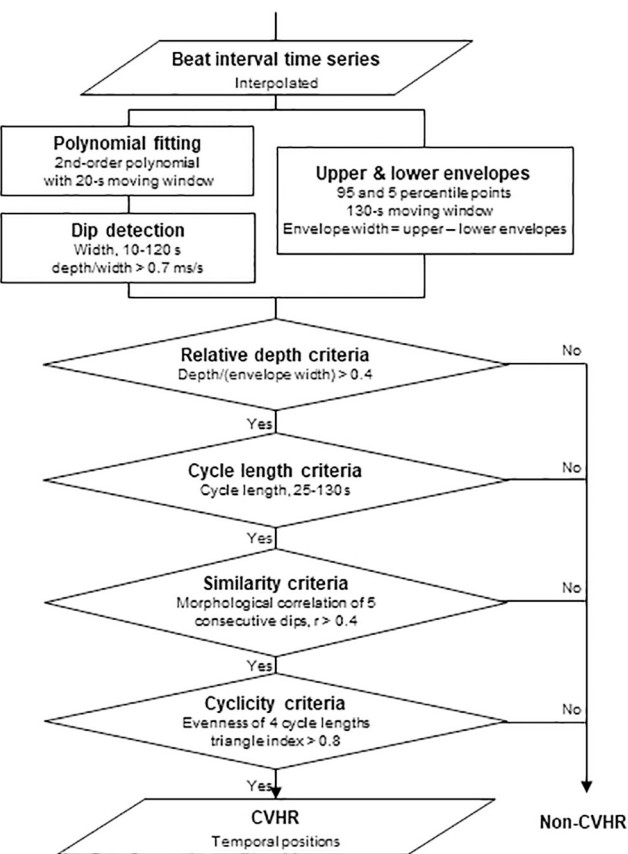

**Fig 1. Algorithm of autocorrelated wave detection with adaptive threshold (ACAT).** The algorithm detects the temporal positions of cyclic variation of heart rate (CVHR) in the beat interval time series as the cyclic and autocorrelated dips that meet four specific criteria (modified from Fig 1 in Ref. [11]).

frequency (0.0033–0.04 Hz) power (VLF), low-frequency (0.04–0.15 Hz) power (LF), high-frequency (0.14–0.40 Hz) power (HF), and LF-to-HF power ratio (LF/HF) of N-N interval variation. The N-N interval of PI was defined as the interval between 400 and 1400 ms with a difference <20% from the local average interval. The N-N interval of RRI was defined as the interval of consecutive beats in sinus rhythm. All these indices were calculated for the entire data length of the PSG study. The total power, ULF, VLF, LF, and HF were calculated as natural-log transformed values.

## Statistical analysis

The program package of Statistical Analysis System (SAS institute, Cary, NC, USA) was used for statistical analyses. Relationships between quantitative variables were evaluated by linear regression analysis and Pearson's correlation coefficients. Differences in quantitative variables between groups dichotomized by AHI level were evaluated by $t$-test. The discriminant performance of indices between dichotomized AHI groups was evaluated by the area under the curve (AUC) of receiver-operating characteristic (ROC) curve. The significance of differences in AUC between indices were evaluated with Hanley and McNeil's method [26]. The discriminatory performance of indices with cutoff thresholds was evaluated with the sensitivity, specificity, accuracy, and positive and negative predictive values (PPV and NPV, respectively). The GLM procedure was used to evaluate the factors that influence the relationship between Fcv

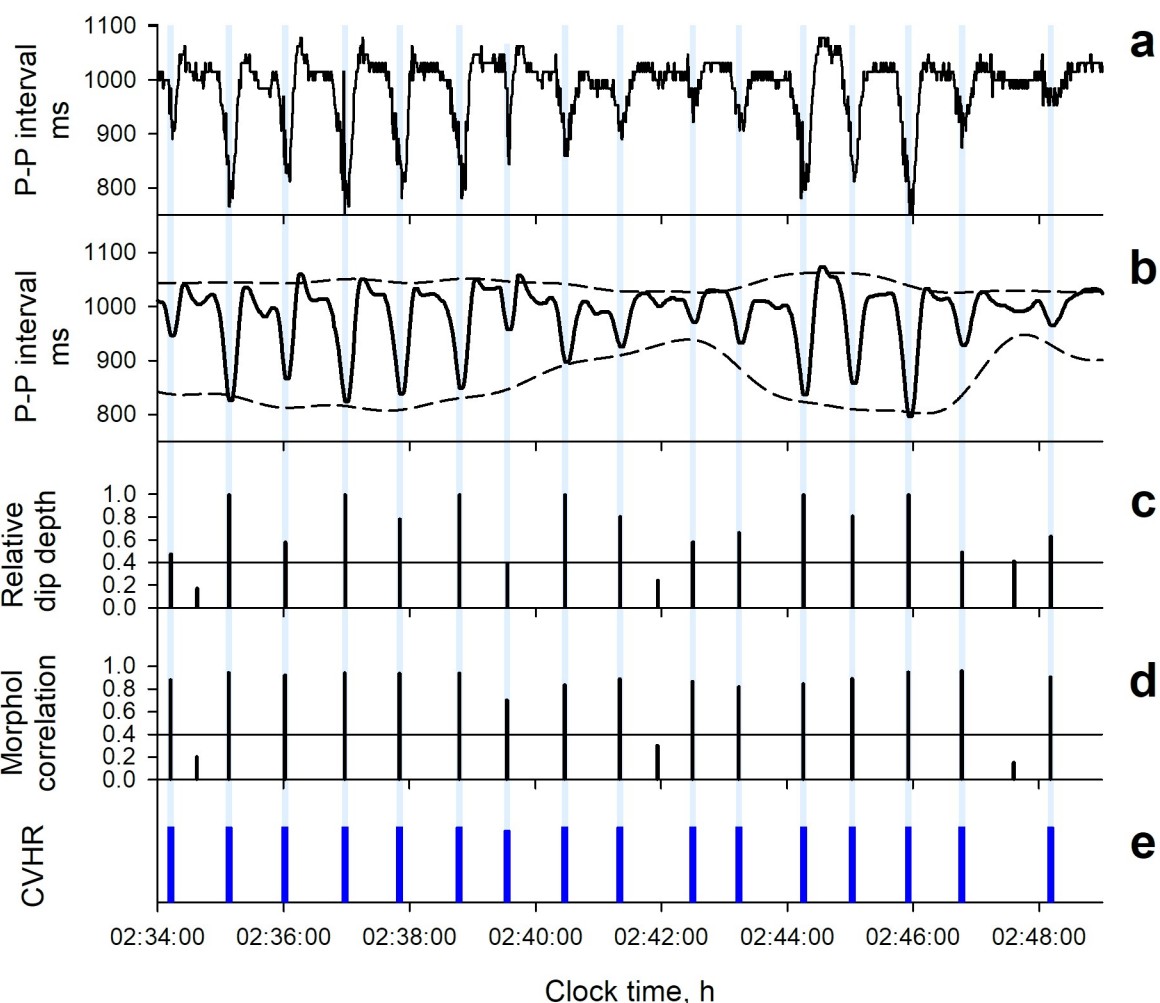

**Fig 2. Detection of CVHR from photoplethysmography (PPG) pulse interval time series by the ACAT algorithm in a representative subject (a 66-y male with a body mass index of 27.0 kg/m² and an apnea-hypopnea index of 79.3).** Panel a: original pulse interval time series. Panel b: second-order polynomial fitting line (solid line) and the upper and lower envelopes of the fitting line (dashed lines). Panel c: the relative dip depth to the envelope width at the time. Panel d: mean morphological correlation coefficients of dip with the two preceding and two subsequent dips. Panel e: temporal positions (blue bars) of dips detected as CVHR.

and AHI, by which examining if the factors had a significant effect on the linear regression model of AHI by Fcv. Statistical significance was considered for $P < 0.05$.

## Results

### Subjects' characteristics

Forty-one consecutive patients (age [IQR], 47 [42–58] year, 7 females) who underwent diagnostic PSG for sleep disordered breathing were studied (Table 1). They had a body mass index (BMI) of 25.4 (IQR, 23.4–28.2) kg/m² and an Epworth Sleepiness Scale score of 4.5 (1.3–16.6). In the PSG study, the median AHI (IQR) was 17.2 (4.4 to 28.4), 22 (54%) of the subjects had an AHI ≥15, and the proportion of obstructive, central, and mixed apnea episodes were 85%, 4%, and 11%, respectively.

**Table 1. Patients' characteristics (n = 41).**

| | |
|---|---|
| Age, y | 48 (42–58) |
| Female (%) | 7 (17%) |
| Body mass index, kg/m$^2$ | 25.4 (23.4–28.2) |
| Systolic blood pressure, mm Hg | 128 (114–139) |
| Diastolic blood pressure, mm Hg | 81 (74–91) |
| Epworth Sleepiness Scale score | 4.5 (1.3–16.6) |
| Time in bed, min | 471 (448–479) |
| Total sleep time, min | 375 (332–418) |
| Sleep efficiency, % | 81.4 (74.4–88.2) |
| AHI | 17.2 (4.4–28.4) |
| AI | 1.5 (0.5–6.9) |
| HI | 12.0 (4.4–16.2) |
| OAI | 1.3 (0.2–6.9) |
| CAI | 0.1 (0.0–0.3) |
| MAI | 0.1 (0.0–0.4) |
| 2% ODI | 15.2 (3.6–26.1) |
| 3% ODI | 8.0 (2.4–20.5) |
| AHI >15 (%) | 22 (54%) |

Data are median (IQR) or number (%).

The denominator in the calculation of the apnea and hypopnea indices is time in bed.

AHI = apnea-hypopnea index; AI = apnea index; HI = hypopnea index; OAI = obstructive apnea index;

CAI = central apnea index; MAI = mixed apnea index; ODA = oxygen desaturation index.

## Detection of CVHR from PPG PI and ECG RRI time series

Fig 3 shows ECG RRI and PPG PI time series simultaneously recorded in a representative subject. Although the two trends differ in the fine structure of fluctuations, the repeated dips corresponding to CVHR (blue bars) appear similarly in both trends. Among indices of PI and RRI variability, those reflecting long-term fluctuations (mean NN interval, SDNN, SDANN, and total power) showed closer correlations between PI and RRI than those reflecting short-term fluctuations (DFA $\alpha_1$, LF, and HF) (Table 2). Fcv measured from PI correlated with Fcv from RRI at $r = 0.90$ (Fig 4b).

## Relationships between PI and RRI variability and sleep apnea indices

Table 3 shows the correlation coefficients of PI and RRI variability indices with sleep apnea indices. Fcv measured from PI correlated with AHI at $r = 0.81$, while Fcv from RRI at $r = 0.85$. Common to both PI and RRI, Fcv showed closer correlations with AI than with HI, and with OAI than with CAI, and it also showed close correlations with 2% and 3% ODIs. No such strong correlations were observed for either time-domain, frequency-domain, or nonlinear indices of variabilities of either PI or RRI, although weak positive correlations were observed between VLF and AI for both PI and RRI.

## Performance of screening for moderate-to-severe sleep apnea

To examine the performance of the Fcv of PI and RRI as indicators for screening sleep apnea, the Fcv was compared between subjects with AHI <15 (no or mild sleep apnea) and those with AHI ≥15 (moderate-to-severe sleep apnea). While no significant group difference was observed for any variability index of either PI or RRI, Fcv was greater in subjects with

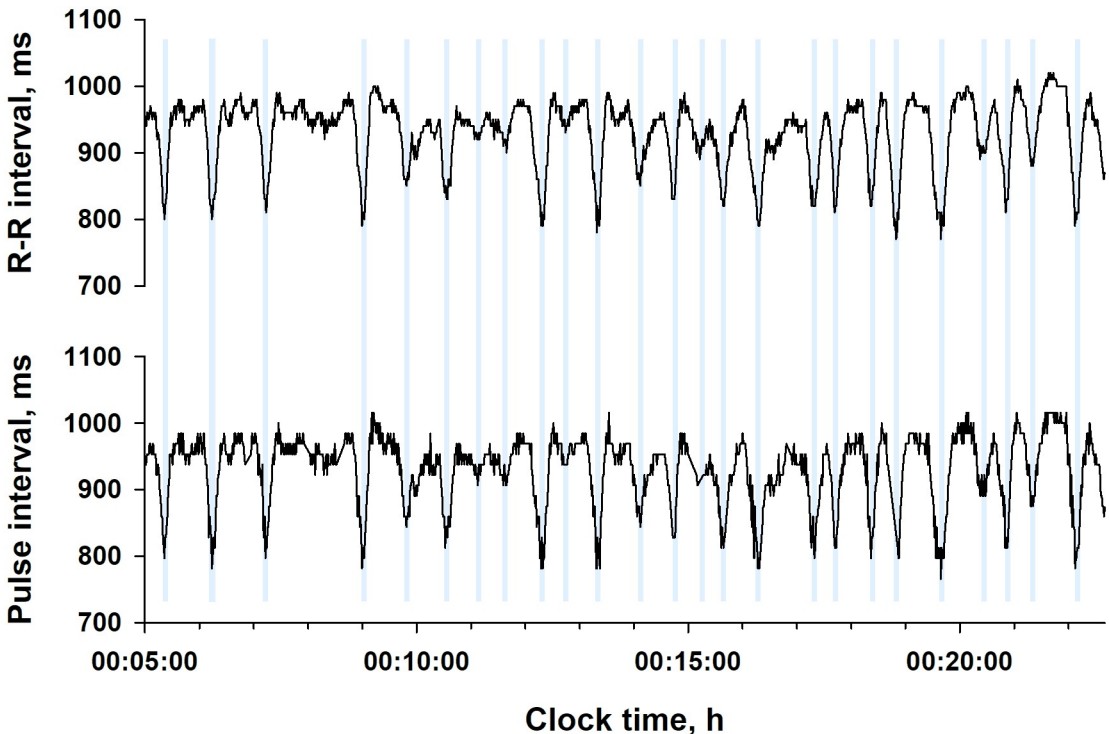

**Fig 3. R-R interval and pulse interval time series from simultaneously recoded electrocardiogram (ECG) and PPG in a representative subject (a 66-y male with an AHI of 79.3).** Vertical blue lines show the temporal positions of CVHR.

moderate-to-severe sleep apnea for both PI and RRI (Table 4). The AUC of ROC curve for the group classification was 0.84 and 0.89 for the Fcv of PI and RRI, respectively, which were significantly greater than the AUC of any other variability indices.

ROC curve analysis revealed that the optimal cutoff of Fcv to screening moderate-to-sever sleep apnea was 11 /h for PI and 15 /h for RRI. With these cutoff values, the Fcv of PI detected moderate-to-sever sleep apnea with 89% specificity, 82% sensitivity, 81% NPV, 90% PPV, and 85% accuracy, and the Fcv of RRI detected it with 95% specificity, 82% sensitivity, 82% NPV, 95% PPV, and 88% accuracy (Table 5). Fig 4 show the linear relationships between the Fcv of PI and RRI and AHI with the cutoff values.

### Factors affecting the relationships between Fcv and AHI

The effects of age, sex, BMI (both continuous and dichotomized at 25 kg/m$^2$), systolic and diastolic blood pressures, Epworth Sleepiness Scale score, time in bed, total sleep time, and sleep efficiency on the relationships of Fcv and AHI were analyzed by the GLM procedure. None of these factors, however, had a significant impact on the relationships for the Fcv of either PI or RRI.

In the case-based analyses, Fcv of PI was <11 /h in four subjects with AHI >15 (false negative) and FCV of PI was >11 /h in two subjects with AHI <15 (false positive). Among the false-negative cases, one subject had frequent ventricular ectopic beats (54% of total beats) during the PSG, while Fcv of ECG was >15 /h and correctly detected CVHR even in this case. In the other three false-negative subjects, ECG Fcv was also <15 /h. Beat interval variations associated with their sleep apnea episodes showed large cycle length variability and were excluded

**Table 2. Correlations between pulse interval and R-R interval variability indices calculated from simultaneously recorded photoplethysmography (PPG) and electrocardiography (ECG) (n = 41).**

|  | Correlation coefficient $r$ |
|---|---|
| Mean NN interval | 0.99 |
| SDNN | 0.96 |
| SDANN | 0.96 |
| rMSSD | 0.74 |
| DC | 0.88 |
| DFA $\alpha_1$ | 0.42 |
| DFA $\alpha_2$ | 0.74 |
| $\beta$ | 0.64 |
| Total power | 0.90 |
| ULF | 0.89 |
| VLF | 0.74 |
| LF | 0.69 |
| HF | 0.64 |
| LF/HF | 0.74 |
| Fcv | 0.90 |

All correlation coefficients are significant.

NN = normal-to-normal; SDNN = standard deviation of NN interval; SDAPI = standard deviation of 5-min average NN intervals; rMSSD = root mean square of successive differences in NN interval; DC = deceleration capacity of NN intervals; DFA = detrended fluctuation analysis; $\alpha_1$ and $\alpha_2$ = short-term (4–11 beats) and long-term (>11 beats) scaling exponents of NN interval fluctuation; $\beta$ = spectral exponent of NN interval fluctuation; ULF = ultra-low-frequency (<0.0033 Hz) power of NN interval variation; VLF = very-low-frequency (0.0033–0.04 Hz) power; LF = low-frequency (0.04–0.15 Hz) power; HF = high-frequency (0.14–0.40 Hz) power; LF/HF = LF-to-HF ratio; Fcv = frequency of cyclic variation.

from CVHR by the cyclicity criteria (Fig 1). In the false-positive cases, non-specific beat interval variations were misidentified as CVHR due to over-adjustment for low CVHR amplitudes during the classification step based on the relative depth criteria (Fig 1). This error did not occur with ECG Fcv.

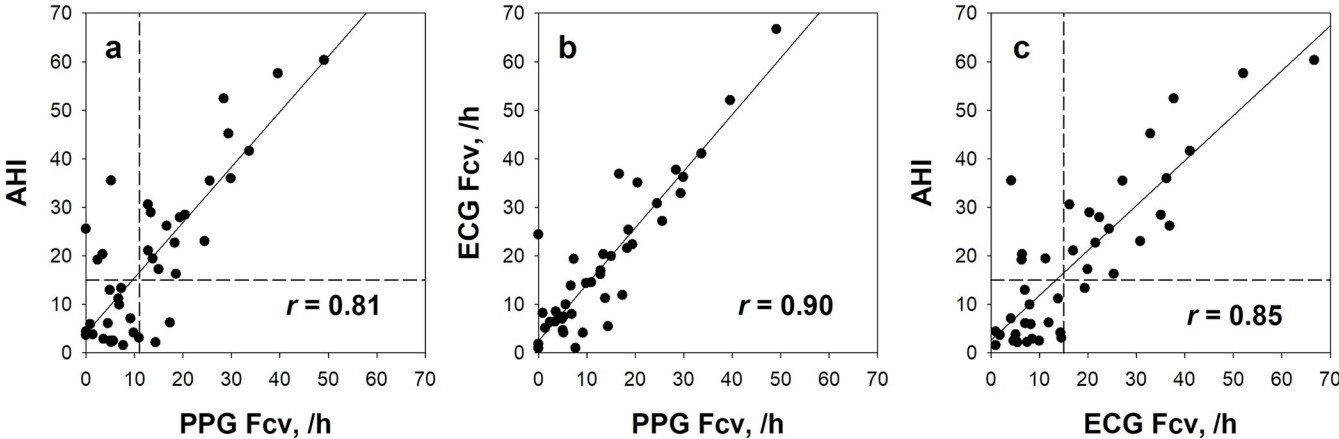

**Fig 4. Relationships of PPG and ECG Fcv with AHI.** In all panels a-c, the plots represent individual subjects. The solid line in each panel represents the linear regression line of the data for all subjects. Horizontal and vertical dashed lines in panels a and c represent the thresholds of 15 for AHI, 11 /h for PPG Fcv, and 15 /h for ECG Fcv, respectively.

**Table 3. Correlations of PPG pulse interval and ECG R-R interval variability indices with sleep apnea indices (n = 41).**

|  | AHI | AI | HI | OAI | CAI | MAI | 2% ODI | 3% ODI |
|---|---|---|---|---|---|---|---|---|
| *PPG pulse interval* | | | | | | | | |
| SDNN | 0.00 | 0.20 | -0.24 | 0.18 | 0.10 | 0.15 | 0.02 | 0.06 |
| SDANN | 0.02 | 0.17 | -0.18 | 0.17 | 0.03 | 0.11 | 0.02 | 0.05 |
| rMSSD | -0.14 | 0.04 | -0.32* | 0.05 | -0.02 | 0.01 | -0.12 | -0.07 |
| DC | 0.02 | 0.08 | -0.07 | 0.05 | 0.19 | 0.12 | 0.02 | 0.04 |
| DFA $\alpha_1$ | 0.17 | 0.27 | -0.02 | 0.22 | 0.26 | 0.31* | 0.17 | 0.18 |
| DFA $\alpha_2$ | -0.22 | -0.32* | -0.01 | -0.31* | -0.15 | -0.22 | -0.23 | -0.26 |
| β | -0.21 | -0.24 | -0.09 | -0.23 | -0.21 | -0.14 | -0.21 | -0.21 |
| Total power | -0.02 | 0.20 | -0.29 | 0.21 | 0.07 | 0.07 | -0.01 | 0.04 |
| ULF | -0.20 | -0.05 | -0.31* | -0.02 | -0.11 | -0.12 | -0.19 | -0.15 |
| VLF | 0.13 | 0.31* | -0.15 | 0.30 | 0.22 | 0.20 | 0.14 | 0.17 |
| LF | 0.04 | 0.21 | -0.18 | 0.19 | 0.25 | 0.16 | 0.05 | 0.08 |
| HF | 0.10 | 0.23 | -0.10 | 0.24 | 0.16 | 0.05 | 0.10 | 0.13 |
| LF/HF | -0.08 | -0.05 | -0.09 | -0.11 | 0.10 | 0.17 | -0.10 | -0.12 |
| Fcv | 0.81* | 0.80* | 0.53* | 0.76* | 0.29 | 0.58* | 0.83* | 0.83* |
| *ECG R-R interval* | | | | | | | | |
| SDNN | 0.00 | 0.18 | -0.24 | 0.17 | 0.10 | 0.14 | 0.01 | 0.04 |
| SDANN | -0.09 | 0.04 | -0.22 | 0.03 | 0.01 | 0.05 | -0.09 | -0.06 |
| rMSSD | 0.06 | 0.27 | -0.23 | 0.30 | 0.02 | 0.06 | 0.08 | 0.11 |
| DC | 0.06 | 0.16 | -0.10 | 0.12 | 0.25 | 0.19 | 0.06 | 0.08 |
| DFA $\alpha_1$ | -0.01 | 0.08 | -0.12 | 0.03 | 0.16 | 0.18 | 0.00 | 0.03 |
| DFA $\alpha_2$ | 0.08 | 0.01 | 0.13 | 0.08 | -0.05 | -0.22 | 0.06 | 0.04 |
| β | -0.02 | -0.17 | 0.18 | -0.16 | -0.12 | -0.10 | -0.03 | -0.05 |
| Total power | 0.07 | 0.29 | -0.25 | 0.30 | 0.13 | 0.13 | 0.08 | 0.12 |
| ULF | -0.11 | 0.05 | -0.26 | 0.08 | -0.01 | -0.07 | -0.10 | -0.07 |
| VLF | 0.19 | 0.40* | -0.16 | 0.38* | 0.24 | 0.28 | 0.21 | 0.25 |
| LF | 0.00 | 0.29 | -0.38* | 0.26 | 0.25 | 0.21 | 0.02 | 0.08 |
| HF | 0.06 | 0.26 | -0.22 | 0.27 | 0.14 | 0.08 | 0.07 | 0.11 |
| LF/HF | -0.05 | -0.03 | -0.06 | -0.10 | 0.11 | 0.18 | -0.07 | -0.10 |
| Fcv | 0.85* | 0.80* | 0.60* | 0.74* | 0.33* | 0.62* | 0.85* | 0.85* |

*Significant correlation coefficients.

The abbreviations are explained in the footnote to Tables 1 and 2.

## Discussions

By applying the ACAT algorithm developed for the automated detection of sleep apnea from ECG, we investigated if sleep apnea can be detected from PPG signals obtained by a wearable watch device. We observed that the Fcv measured from PPG correlated with AHI at $r = 0.81$ and Fcv values $\geq 11$ /h detected subjects with moderate-to-severe sleep apnea (AHI $\geq 15$) with 89% specificity, 82% sensitivity, 81% NPV, 90% PPV, and 85% accuracy. The classification performance of the Fcv of PPG was comparable to that of ECG. Our findings indicate that by using the ACAT algorithm, PPG signal obtained from a wearable watch device can be used to screen patients with moderate-to-severe sleep apnea.

To our knowledge, this is the first study to show that sleep apnea episodes in adults can be quantitatively detected by PPG pulse wave signals of wearable watch devices. In a study of simultaneous recording of PPG and ECG in normal subjects and patients with obstructive sleep apnea during sleep, Khandoker et al [27] analyzed pulse rate and heart rate variability.

**Table 4. PPG pulse interval and ECG R-R interval variability indices in patients grouped by AHI and the discriminant performance of the indices.**

| | AHI ≤15 (n = 19) | AHI >15 (n = 22) | P (t-test) | AUC |
|---|---|---|---|---|
| *PPG pulse interval* | | | | |
| SDNN, ms | 86 ± 35 | 77 ± 26 | 0.3 | 0.54* |
| SDANN, ms | 58 ± 26 | 54 ± 21 | 0.5 | 0.54* |
| rMSSD, ms | 53 ± 20 | 43 ± 18 | 0.1 | 0.68 |
| DC, ms | 8.1 ± 2.4 | 7.8 ± 3 | 0.7 | 0.50* |
| DFA $\alpha_1$ | 0.93 ± 0.13 | 0.95 ± 0.23 | 0.8 | 0.54* |
| DFA $\alpha_2$ | 1.01 ± 0.07 | 1.00 ± 0.13 | 0.7 | 0.58* |
| β | 1.19 ± 0.14 | 1.15 ± 0.22 | 0.5 | 0.62* |
| Total power, ms$^2$ | 8.7 ± 0.6 | 8.5 ± 0.8 | 0.2 | 0.58* |
| ULF, ms$^2$ | 8.2 ± 0.8 | 7.7 ± 0.9 | 0.06 | 0.66 |
| VLF, ms$^2$ | 7.2 ± 0.5 | 7.0 ± 1.2 | 0.5 | 0.47* |
| LF, ms$^2$ | 5.9 ± 0.7 | 5.5 ± 1.4 | 0.2 | 0.53* |
| HF, ms$^2$ | 5.7 ± 0.8 | 5.5 ± 1.6 | 0.5 | 0.46* |
| LF/HF | 1.4 ± 0.7 | 1.3 ± 0.8 | 0.5 | 0.59* |
| Fcv, /h | 6.4 ± 4.6 | 19.6 ± 12.3 | <0.0001 | 0.84 |
| *ECG R-R interval* | | | | |
| SDNN, ms | 94 ± 43 | 85 ± 29 | 0.4 | 0.53† |
| SDANN, ms | 67 ± 42 | 55 ± 21 | 0.3 | 0.55† |
| rMSSD, ms | 39 ± 22 | 37 ± 21 | 0.7 | 0.51† |
| DC, ms | 9.8 ± 2.1 | 9.3 ± 3.3 | 0.6 | 0.50† |
| DFA $\alpha_1$ | 1.13 ± 0.28 | 1.07 ± 0.43 | 0.6 | 0.53† |
| DFA $\alpha_2$ | 1.02 ± 0.08 | 1.08 ± 0.15 | 0.1 | 0.62† |
| β | 1.13 ± 0.15 | 1.19 ± 0.16 | 0.2 | 0.61† |
| Total power, ms$^2$ | 8.7 ± 0.5 | 8.6 ± 0.8 | 0.5 | 0.53† |
| ULF, ms$^2$ | 7.9 ± 0.6 | 7.7 ± 0.9 | 0.4 | 0.55† |
| VLF, ms$^2$ | 7.6 ± 0.7 | 7.6 ± 0.9 | 0.8 | 0.50† |
| LF, ms$^2$ | 6.4 ± 0.6 | 5.9 ± 1.1 | 0.06 | 0.61† |
| HF, ms$^2$ | 5.7 ± 1.1 | 5.5 ± 1.2 | 0.4 | 0.50† |
| LF/HF | 2.52 ± 1.7 | 1.87 ± 1.15 | 0.1 | 0.60† |
| Fcv, /h | 8.1 ± 5 | 26.9 ± 15.2 | <0.0001 | 0.89 |

*Significantly smaller than the AUC of PPG Fcv.

†Significantly smaller than the AUC of ECG Fcv.

AUC = area under the receiver-operating curve for classification between patients with AHI ≤15 and >15. The other abbreviations are explained in the footnotes to Tables 1 and 2.

**Table 5. Classification performance of PPG and ECG Fcv between patients grouped by AHI.**

| | | AHI ≤15 | AHI >15 | |
|---|---|---|---|---|
| *PPG pulse interval* | Fcv <11/h | 17 | 4 | NPV = 81% |
| | Fcv ≥11/h | 2 | 18 | PPV = 90% |
| | | Specificity = 89% | Sensitivity = 82% | Accuracy = 85% |
| *ECG R-R interval* | Fcv <15/h | 18 | 4 | NPV = 82% |
| | Fcv ≥15/h | 1 | 18 | PPV = 95% |
| | | Specificity = 95% | Sensitivity = 82% | Accuracy = 88% |

NPV = negative predictive value; PPV = positive predictive value. The other abbreviations are explained in the footnotes to Tables 1 and 2.

They found that frequency domains and complexity analysis measures of both pulse rate and heart rate variability differ between 2-min epochs of normal and sleep apnea events. They also observed that several variability measures (SDNN, rMSSD, HF, LF/HF, and sample entropy) of pulse rate and heart rate differ significantly during sleep apnea. They concluded that pulse rate variability could discriminate between epochs with and without sleep apnea, but it does not precisely reflect heart rate variability in sleep disordered breathing. Using the Apple Watch device, Tison et al [28] collected PPG pulse rate and step count continuously for an average of 8.9 weeks in 6,115 subjects. They performed machine learning using deep neural networks and reported that the developed model predicted prevalent sleep apnea with 90.4% sensitivity and 59.8% specificity. The present study is in the same line with these earlier studies, but none of the earlier studies has reported the quantitative relationships of metrics derived from PPG with AHI. The method we reported here detects each episode of sleep apnea as the CVHR of PI and thus, it can estimate the number and temporal positions of sleep apnea episodes.

The ACAT algorithm we used to detect CVHR has been developed and optimized for ECG RRI [11–13], but the present study indicated that the ACAT algorithm can also be used for PPG PI and performs as well as for ECG RRI. Many studies have reported discrepancies and non-substitution of pulse rate and heart rate variability [15–19], particularly in various diseases including sleep apnea [27]. Nonetheless, the present study showed equivalence of PPG and ECG in the detection of sleep apnea. This seems to be due to the characteristics of CVHR and to the features of ACAT algorithm. Although inconsistencies in pulse rate and heart rate variability have been reported with indicators that quantify short-term fluctuations such as LF and HF components [8, 16, 19], CVHR is a long-term fluctuation with a cycle length between 25 and 130 s. In fact, we observed almost the same wave forms of CVHR for ECG RRI and PPG PI, while higher-frequency fluctuations showed apparent differences (Fig 3). Consistent with this, closer correlations between RRI and PI were observed for indices reflecting long-term fluctuations than for those reflecting short-term fluctuations (Table 2). Additionally, the ACAT algorithm performs the preprocessing of PI signals by 2nd-order polynomial fitting with 20-s moving window to remove high frequency fluctuations with a cycle length $<\sim$20 s (Figs 1 and 2). Furthermore, the algorithm uses an adaptive threshold (depth relative to the envelope range) to detect dips, rather than a fixed threshold. These features are considered to enhance robustness of the algorithm against differences in short-term components and in amplitudes between heart rate and pulse rate variability.

For both PI and RRI, Fcv closely correlated with 2% and 3% ODI, and showed closer correlations with AI than with HI (Table 3). These suggest that CVHR may be more likely to occur with apnea causing hypoxemia than with hypopnea. In an earlier study, Nakano et al. [29] reported the effect of BMI on the relationship between ODI and AHI. They reported that 3% ODI ≥15 was the optimal cutoff to detect moderate-to-severe sleep apnea (AHI ≥15) in patients with BMI ≥25 kg/m$^2$, while 3% ODI ≥10 was the optimal cutoff in patients with BMI <25 kg/m$^2$, suggesting that sleep apnea in obese patients has a greater apnea-to-hypopnea ratio than that in non-obese patients. In the present study, we observed no significant impact of BMI on the relationship between Fcv and AHI; but we observed no significant difference either in apnea-to-hypopnea ratio between those with BMI ≥25 kg/m$^2$ and <25 kg/m$^2$ among patients with AHI ≥15 (apnea-to-hypopnea ratio, 1.3 ± 1.4 vs. 0.4 ± 0.3, $P$ = 0.07). Further studies are needed to clarify the impacts of obesity on the estimation of AHI by Fcv.

Given the widespread availability of wearable watch devices with a PPG sensor, the results of the present study show great potential for using this social resource as a cost-effective large-scale screening of sleep apnea. Sleep apnea is an increased risk of traffic accidents [3, 4] and cardiovascular diseases [5], and this risk may be reduced by treatment [30–32]. Finding patients who have been left undiagnosed will have great benefits to social safety and the health

care economy. Currently, many excellent portable devices are available for sleep apnea screening, but their use depends on the patient's motivation. Even if the diagnostic accuracy of PPG sleep apnea detection is comparable to other Type 4 portable devices [33], PPG wearable devices used regardless of motivation may contribute to the detection of sleep apnea in previously unscreened populations.

This study has several limitations. First, we used only one type of wearable watch device (E4 wristband, Empatica, Milano, Italy) and the pulse wave signals were analyzed by the manufacturer's cloud application via Internet (E4 connect, Empatica, Milan, Italy). Therefore, the results may vary when using other types of device and when measuring PI in other ways. The ACAT algorithm, however, uses only smoothed PI data (2nd-order polynomial fitting with 20-s moving window), which makes the algorithm robust against noises and variations in the accuracy of PI measurement. Second, the ACAT algorithm was developed to detect sleep apnea from sinus rhythm ECG and is not applicable to atrial fibrillation. We excluded subjects who showed continuous atrial fibrillation during the PSG, but sleep apnea, particularly its central form, often accompanies atrial fibrillation [5]. Development of sleep apnea detection algorithms under atrial fibrillation is desired. Third, this study compared the Fcv frequency with AHI whose denominator was time in bed. This is because the PPG method cannot estimate total sleep time (time in bed minus time of awakening). As a result, the Fcv frequency can underestimate AHI with total sleep time as the denominator. To compensate for this difference, a method of estimating sleep efficiency from wearable devices need to be incorporated. Finally, this study used the ACAT algorithm to detect CVHR by PPG, but the classification criteria of this algorithm are optimized for ECG detection of CVHR. The performance of CVHR detection by PPG may be improved by PPG-specific classification criteria. These limitations are topics for future studies.

## Conclusions

We examined if sleep apnea can be quantitatively detected from PPG pulse intervals by an automated algorithm called ACAT. The Fcv of PI measured by the ACAT correlated with AHI at $r = 0.81$ and Fcv values $\geq 11$ /h detected subjects with moderate-to-severe sleep apnea (AHI $\geq 15$) with a PPV of 90%. The analysis of nighttime PPG by the ACAT algorithm could be used for the quantitative screening for sleep apnea.

## Acknowledgments

The ACAT algorithm is implemented in a Suzuken Holter ECG scanner (Cardy Analyzer 05, Suzuken Co., Ltd., Nagoya, Japan) and is commercially available.

## Author Contributions

**Conceptualization:** Junichiro Hayano, Makoto Komazawa, Kenichi Itao, Haruhito Tanaka, Emi Yuda.

**Data curation:** Makoto Komazawa, Norihiro Ueda.

**Formal analysis:** Junichiro Hayano.

**Funding acquisition:** Junichiro Hayano, Kenichi Itao, Emi Yuda.

**Investigation:** Hiroaki Yamamoto, Izumi Nonaka, Emi Yuda.

**Methodology:** Junichiro Hayano, Hiroaki Yamamoto, Izumi Nonaka, Makoto Komazawa, Norihiro Ueda.

**Project administration:** Junichiro Hayano, Kenichi Itao, Haruhito Tanaka, Emi Yuda.

**Resources:** Makoto Komazawa, Kenichi Itao, Norihiro Ueda, Haruhito Tanaka, Emi Yuda.

**Software:** Junichiro Hayano, Makoto Komazawa, Kenichi Itao.

**Supervision:** Kenichi Itao, Haruhito Tanaka, Emi Yuda.

**Validation:** Hiroaki Yamamoto, Izumi Nonaka, Kenichi Itao, Norihiro Ueda, Haruhito Tanaka.

**Visualization:** Hiroaki Yamamoto.

**Writing – original draft:** Junichiro Hayano.

**Writing – review & editing:** Hiroaki Yamamoto, Norihiro Ueda, Haruhito Tanaka, Emi Yuda.

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
