## [Decision Letter · Decision Letter 0]

11 Sep 2020

PONE-D-20-22289

Quantitative detection of sleep apnea with wearable watch device

PLOS ONE

Dear Dr. Hayano,

Thank you for submitting your manuscript to PLOS ONE. After careful consideration, we feel that it has merit but does not fully meet PLOS ONE’s publication criteria as it currently stands. Therefore, we invite you to submit a revised version of the manuscript that addresses the points raised during the review process.

The manuscript describes an interesting and novel approach to screening for sleep apnea, with clinical relevance. The methods are sound and the manuscript is well written. With appropriate revisions that address the reviewers comments, this manuscript is a good candidate for publication.

We look forward to receiving your revised manuscript.

Kind regards,

Allan R. Martin

Academic Editor

PLOS ONE

Journal Requirements:

2.We note that you have indicated that data from this study are available upon request. PLOS only allows data to be available upon request if there are legal or ethical restrictions on sharing data publicly. For information on unacceptable data access restrictions, please see http://journals.plos.org/plosone/s/data-availability#loc-unacceptable-data-access-restrictions.

[The authors have declared that no competing interests exist.].   

We note that one or more of the authors are employed by a commercial company: WINFrontier Co., Ltd.,

Additional Editor Comments (if provided):

The manuscript describes an interesting and novel approach to screening for sleep apnea, with clinical relevance. The methods are sound and the manuscript is well written. I recommend that the manuscript is accepted after minor revisions.

Reviewers' comments:

Reviewer's Responses to Questions

**Comments to the Author**

1. Is the manuscript technically sound, and do the data support the conclusions?

Reviewer #1: Yes

Reviewer #2: Yes

2. Has the statistical analysis been performed appropriately and rigorously? 

Reviewer #1: Yes

Reviewer #2: Yes

3. Have the authors made all data underlying the findings in their manuscript fully available?

Reviewer #1: Yes

Reviewer #2: Yes

4. Is the manuscript presented in an intelligible fashion and written in standard English?

Reviewer #1: Yes

Reviewer #2: Yes

5. Review Comments to the Author

Reviewer #1: Dear authors,

This article presented an interesting new method of OSAS wide screening based on cardiac activity.

I have only minor comments to formulate.

Please reduce the occurrence of acronyms to improve the understanding of your text.

Short title could be improved (Sleep apnea detection by photoplethismography for example)

l.43-44: add 'sleepiness, reduced sleep quality between 'with' and 'reduced labor...'Please revise the reference if necessary.

l.103-104: The choice of TIB for AHI calculation is unusual with PSG. It leads to an underestimate of the severity of OSAS like with ventolatory polygraphy.

l.175-176 and l. 246-247: the sentences are not clear and need a reformulation.

l.288-290: PPV would be more appropriate. So, you could replace '89%... accuracy' by '90% positive predictive value'

In the discussion and the conclusion, it may be fuitful to compare the efficiency of your method of detection to other ones.

I am thinking particularly about automatic one-channel polygraphy (RU SLEEPING for example). In other words, your method seems to be as efficient as previously mentionned method.

Reviewer #2: This study by Hayano et al. was designed to investigate if PPG signal data could be used to detect sleep apnea.

The authors already established the screening method for sleep apnea by cyclic variations of heart rate (CVHR) from ECG r-r interval using ACAT algorithm.

Hayano J, Watanabe E, Saito Y, Sasaki F, Fujimoto K, et al. (2011) Screening for obstructive sleep apnea by cyclic variation of heart rate. Circ Arrhythm Electrophysiol 4: 64-72.

Hayano J, Tsukahara T, Watanabe E, Sasaki F, Kawai 328 K, et al. (2013) Accuracy of ECG based screening for sleep-disordered breathing: a survey of all male workers in a transport company. Sleep Breath 17: 243-251.

The authors investigated the diagnostic value of CVHR from PPG pulse rate (PPG-Fcv) and concluded it could be used for the quantitative screening for sleep apnea. It is interesting topic, I have just a few comments.

1. I think you cannot evaluate PPG-Fcv if the patients have atrial fibrillation or other arrhythmic disease. Atrial fibrillation is sometimes accompanied by SAS, you should mention this problem as a limitation.

2. In figure 4-a, there are four patients with high-AHI and low-PPG Fcv (false-negative), and two patients with low-AHI and high-PPG Fcv (false-positive). It may not be a big problem in number, please add more discussion as to the main cause of false positive and false negative.

6. PLOS authors have the option to publish the peer review history of their article (what does this mean?). If published, this will include your full peer review and any attached files.

Reviewer #1: **Yes: **Dr COSTE Olivier (MD, PhD) - Lyon (France)

Reviewer #2: No

---

## [Author Response · Author response to Decision Letter 0]

21 Sep 2020

Reviewer #1: Dear authors,

This article presented an interesting new method of OSAS wide screening based on cardiac activity.

I have only minor comments to formulate.

Please reduce the occurrence of acronyms to improve the understanding of your text.

[Response] We removed unnecessary acronyms, thank you.

Short title could be improved (Sleep apnea detection by photoplethismography for example)

[Response] We revised short title to “Sleep apnea detection by photoplethysmography” as recommended.

l.43-44: add 'sleepiness, reduced sleep quality between 'with' and 'reduced labor...'Please revise the reference if necessary.

[Response] We revised as indicated, thank you (L46).

l.103-104: The choice of TIB for AHI calculation is unusual with PSG. It leads to an underestimate of the severity of OSAS like with ventolatory polygraphy.

[Response] The PPG method cannot estimate TST because the duration of awakening during TIB cannot be determined. Therefore, the frequency of Fcv detected in PPG can only be expected to correlate with AHI whose denominator was time in bed. Consequently, Fcv estimated by PPG can under-estimate the AHI with TST as the denominator. To adjust this difference, a method of estimating sleep efficiency (TST-to-TIB ratio) from wearable devices need to be incorporated. We added this point to study limitation (L378-382).

l.175-176 and l. 246-247: the sentences are not clear and need a reformulation.

[Response] We revised these sentences, thank you (L203-204 and L331-334).

l.288-290: PPV would be more appropriate. So, you could replace '89%... accuracy' by '90% positive predictive value'

[Response] We revised as indicated, thank you (393).

In the discussion and the conclusion, it may be fuitful to compare the efficiency of your method of detection to other ones.

I am thinking particularly about automatic one-channel polygraphy (RU SLEEPING for example). In other words, your method seems to be as efficient as previously mentionned method.

[Response] The diagnostic accuracy of PPG sleep apnea detection is comparable to other Type 4 devices. However, the use of portable devices depends on patient’s motivation. PPG wearable devices used regardless of motivation may contribute to the detection of sleep apnea in previously unscreened populations. We added this to discussion (L362-366).

Reviewer #2: This study by Hayano et al. was designed to investigate if PPG signal data could be used to detect sleep apnea.

The authors already established the screening method for sleep apnea by cyclic variations of heart rate (CVHR) from ECG r-r interval using ACAT algorithm.

Hayano J, Watanabe E, Saito Y, Sasaki F, Fujimoto K, et al. (2011) Screening for obstructive sleep apnea by cyclic variation of heart rate. Circ Arrhythm Electrophysiol 4: 64-72.

Hayano J, Tsukahara T, Watanabe E, Sasaki F, Kawai 328 K, et al. (2013) Accuracy of ECG based screening for sleep-disordered breathing: a survey of all male workers in a transport company. Sleep Breath 17: 243-251.

The authors investigated the diagnostic value of CVHR from PPG pulse rate (PPG-Fcv) and concluded it could be used for the quantitative screening for sleep apnea. It is interesting topic, I have just a few comments.

1. I think you cannot evaluate PPG-Fcv if the patients have atrial fibrillation or other arrhythmic disease. Atrial fibrillation is sometimes accompanied by SAS, you should mention this problem as a limitation.

[Response] We added the following description to limitations (L374-378): The ACAT algorithm was developed to detect sleep apnea from sinus rhythm ECG and is not applicable to atrial fibrillation. We excluded subjects who showed continuous atrial fibrillation during the PSG, but sleep apnea, particularly its central form, often accompanies atrial fibrillation [5]. Development of a sleep apnea detection algorithm by PPG under atrial fibrillation is desired.

2. In figure 4-a, there are four patients with high-AHI and low-PPG Fcv (false-negative), and two patients with low-AHI and high-PPG Fcv (false-positive). It may not be a big problem in number, please add more discussion as to the main cause of false positive and false negative.

[Response] We added a paragraph for the causes of classification errors (L288-297) and also added sentences to limitation (L382-385).

In the case-based analyses, Fcv of PI was <11/h in four subjects with AHI >15/h (false negative) and FCV of PI was >11/h in two subjects with AHI <15/h (false positive) (Fig 4a). Among the false-negative cases, one subject had frequent ventricular ectopic beats (54% of total beats) during the PSG, while Fcv of ECG detected CVHR correctly even in this case. In the other three false-negative subjects, ECG Fcv was also <15/h. Beat interval variations associated with their sleep apnea episodes showed large cycle length variability and were excluded from CVHR by the cyclicity criteria (Fig 1). In the false-positive cases, non-specific beat interval variations were misidentified as CVHR due to over-adjustment for low CVHR amplitudes during the classification step based on the relative depth criteria (Fig 1). This error did not occur with ECG Fcv.

Limitations

Finally, this study used the ACAT algorithm to detect CVHR by PPG, but the classification criteria of this algorithm are optimized for ECG detection of CVHR. The performance of CVHR detection by PPG may be improved by PPG-specific classification criteria and is a topic for future studies.

---

## [Decision Letter · Decision Letter 1]

22 Oct 2020

Quantitative detection of sleep apnea with wearable watch device

PONE-D-20-22289R1

Dear Dr. Hayano,

We’re pleased to inform you that your manuscript has been judged scientifically suitable for publication and will be formally accepted for publication once it meets all outstanding technical requirements.

Kind regards,

Allan R. Martin

Academic Editor

PLOS ONE

Additional Editor Comments (optional):

Reviewers' comments:

Reviewer's Responses to Questions

**Comments to the Author**

1. If the authors have adequately addressed your comments raised in a previous round of review and you feel that this manuscript is now acceptable for publication, you may indicate that here to bypass the “Comments to the Author” section, enter your conflict of interest statement in the “Confidential to Editor” section, and submit your "Accept" recommendation.

Reviewer #1: All comments have been addressed

Reviewer #2: All comments have been addressed

2. Is the manuscript technically sound, and do the data support the conclusions?

Reviewer #1: Yes

Reviewer #2: Yes

3. Has the statistical analysis been performed appropriately and rigorously? 

Reviewer #1: Yes

Reviewer #2: Yes

4. Have the authors made all data underlying the findings in their manuscript fully available?

Reviewer #1: Yes

Reviewer #2: Yes

5. Is the manuscript presented in an intelligible fashion and written in standard English?

Reviewer #1: Yes

Reviewer #2: Yes

6. Review Comments to the Author

Reviewer #1: The authors have carefullly answered to my requirements during the revision process.

Therefore, this manuscript is now acceptable for publication.

Reviewer #2: The authors improved the manuscript according to the Reviewer's comments, and my comments have been addressed appropriately.

7. PLOS authors have the option to publish the peer review history of their article (what does this mean?). If published, this will include your full peer review and any attached files.

Reviewer #1: **Yes: **Olivier COSTE

Reviewer #2: No

---

## [Editor Report · Acceptance letter]

28 Oct 2020

PONE-D-20-22289R1 

Quantitative detection of sleep apnea with wearable watch device 

Dear Dr. Hayano:

I'm pleased to inform you that your manuscript has been deemed suitable for publication in PLOS ONE. Congratulations! Your manuscript is now with our production department. 

Kind regards, 

on behalf of

Dr. Allan R. Martin 

Academic Editor

PLOS ONE